# Grapevine Plants Management Using Natural Extracts and Phytosynthesized Silver Nanoparticles

**DOI:** 10.3390/ma15228188

**Published:** 2022-11-18

**Authors:** Diana Elena Vizitiu, Daniela Ionela Sardarescu, Irina Fierascu, Radu Claudiu Fierascu, Liliana Cristina Soare, Camelia Ungureanu, Elena Cocuta Buciumeanu, Ionela Catalina Guta, Letitia Mariana Pandelea

**Affiliations:** 1The National Institute for Research & Development for Biotechnology in Horticulture Stefanesti, 110134 Stefanesti, Romania; 2Faculty of Chemical Engineering and Biotechnologies, University “Politehnica” of Bucharest, Bucharest, 313 Splaiul Independentei Str., 060042 Bucharest, Romania; 3The National Institute for Research & Development in Chemistry and Petrochemistry, ICECHIM, 202 Spl. Independentei, 060021 Bucharest, Romania; 4Faculty of Horticulture, University of Agronomic Sciences and Veterinary Medicine of Bucharest, 011464 Bucharest, Romania; 5Natural Science Department, Faculty of Sciences, Physical Education and Informatics, University of Pitesti, 110040 Pitesti, Romania

**Keywords:** *Vitis* spp., *Dryopteris filix-mas*, *Uncinula necator*, pith/wood, natural treatments

## Abstract

Starting from the well-known antimicrobial properties of silver nanoparticles, the goal of this study is to evaluate the influence of two “green” recipes, namely an alcoholic extract of *Dryopteris filix-mas* (L.) Schott and a dispersion of silver nanoparticles phytosynthesized using the extract on grapevine pathogens. The influence of some grapevine parameters (pith/wood rapport, soluble sugars, starch, total sugars, total water content, length of young shoots, number of grapes) in field experiments was also studied. The study was conducted on four clones (Feteasca alba 97 St., Feteasca neagra 6 St., Feteasca regala 72 St., and Cabernet Sauvignon 131 St.) located in vegetation pots inside a greenhouse. For the phytosynthesis of the silver nanoparticles (AgNPs) we used a scaled-up technology, allowing us to obtain large quantities of nanoparticles-containing solution. The AgNPs analysis by X-ray diffraction and transmission electron microscopy confirmed the synthesis of spherical and quasi-spherical nanoparticles of 17 nm average diameter and 6.72 nm crystallite size. The field experiments registered different responses of the four clones to the treatment, using both the natural extracts and phytosynthesized nanoparticles solution. Both recipes exhibited a protective effect against the *Uncinula necator* pathogen. For the treatment using phytosynthesized nanoparticles, significant increases in the pith/wood ratio for white wine clones (Feteasca alba 97 St. and Feteasca regala 72 St.) were observed. The biochemical analyses revealed other significant increases of soluble sugars (red wine clones—Feteasca neagra and Cabernet Sauvignon/second year), starch (Feteasca alba and Cabernet Sauvignon in 2021 for both clones), total sugars (Feteasca alba and Feteasca neagra in 2021 for both clones), and of total water content (Feteasca alba and Feteasca neagra in 2021 for both clones), respectively. The applied treatments also led to an increase of young shoots length and grape numbers for all clones as compared to the control (chemical pesticide), which would suggest a potential biostimulant effect of the recipes.

## 1. Introduction

Viticulture plays an important global role, both from an economic and social point of view. Unfortunately, the vineyards are frequently attacked by numerous pathogens (fungus, viruses, viroids, phytoplasmas, bacteria) that, in a worst-case scenario, can cause serious epidemics, producing significant economic losses and even completely compromising the production [1,2,3].

There are several diseases that develop on grapevines during the different phases of vegetation. Their successful management is ensured by an integrated protection program implementation, including several components, i.e., biological, agrophytotechnical, mechanical, physical, and chemical methods [4,5].

Powdery mildew and downy mildew are among the most important diseases of grapevine [6,7]. According to literature data, powdery mildew (produced by the pathogen *Uncinula necator*) is the oldest grapevine disease known to science and practice. It is native to North America, where the disease has always existed [8,9]. The pathogen is a holoparasite of the *Vitaceae* family (including *Vitis*, *Cissus*, *Parthenocissus* and *Ampelopsis* [10]), and can infect all the green parts of the plants.

Downy mildew, produced by the heterotrophic oomycete *Plasmopara viticola*, is considered one of the most devastating diseases of grapevines, especially in regions with warm and humid growing seasons [11], representing an important limiting factor for the grapevine’s cultivation [12,13].

Phytosanitary treatments against diseases are generally performed using pesticides. However, the latest discoveries in epidemiology and toxicology reveal that pesticides can negatively affect human health by the development of some diseases such as cancer, neurological, and reproductive disorders [14]. As such, considering the world-wide regulations regarding the restriction of several chemical pesticides, there is a high demand for eco-friendly alternatives to their use in order to control the pathogens and to mitigate their negative effects [13].

A viable alternative is represented by the vegetal extracts [15] obtained from different plant species which are not toxic to grapevines and humans [16,17,18,19,20]. For example, successful studies against grapevine pathogens were performed using different plants, such as hibiscus [21], garlic and clove [22], magnolia-bark [23], or even grapevine-canes [24] and grape-stalks [25].

A much lesser studied group of plants is represented by the ferns, which could provide very important resources for the potent antimicrobial recipe development. A possible application would be in the pathogen control area, among others [26,27,28]. Among the existing fern species, the *Dryopteris filix-mas* (L.) Schott. was selected for the present study. The *D. filix-mas* fern can reach a height of 1.2 m [29], being considered a wild vegetable in some countries [30]. According to the available literature data, this fern species contains important secondary metabolites like saponin, tannin, phenol, phytosterol, alkaloids [31], glycoside, terpenoid, and protein [32]. It also contains aromatic, polyketide, monoterpenic, and sesquiterpenic compounds, as well as carotenoid derivatives [33]. Several studies regarding the applications of *D. filix-mas* extracts have been previously published, including the evaluation of the antimicrobial properties of methanol and flavonoid extracts of *D. filix-mas* (alongside *Marchantia polymorpha* L. and *Ephedra foliata* Boiss) against *Alternaria solani, Fusarium oxysporum*, and *Rhizoctonia solani* pathogens. The *D. filix-mas* and *E. foliata* extracts were found to inhibit up to 80% of the *A. solani* mycelium, while *M. polymorpha* and *D. filix-mas* (rhizome extract) completely inhibited the mycelial growth of *R. solani* [34]. Other researchers used aqueous extracts obtained from different parts of *D. filix-mas* (roots and rhizomes) for the control of *Corcyra cephalonica* (Staint.) larvae, obtaining 100% larval mortality [35].

Ethanol and aqueous extracts of *D. filix-mas* were also used to treat foliar fungal diseases *Pestalotiopsis theae* (Saw.) Stey., *Colletotrichum camelliae* Mess., *Curvularia eragrostidis* (P. Hennings) Meyer, and *Botryodiplodia theobromae* (Pat.). The results showed an effective inhibition of spore germination for the two extracts [36]. The ferns *D. filix-mas* and *D. affinis* could be used as ingredients in the cosmetics industry, as preservatives in food, or in antimicrobial therapy [37], and could have a prospective use of natural extracts with antioxidant and antimicrobial natural extracts in general [38]. Our group previously demonstrated the potential of *D. filix-mas* extracts to inhibit the *Venturia inaequalis* and the *Podosphaera leucotricha* (lines responsible for common apple diseases) growth [39]. The antimicrobial and antifungal properties of the hydroalcoholic leaves extracts of another fern, *Asplenium scolopendrium* L., have also been proved [40]. 

The selection of *D. filix-mas* extract for developing such phytosynthesized nanomaterials and for the application in field experiments of the resulting solutions was made considering several literature data inputs. First of all, the well-known antimicrobial potential of the metallic nanoparticles (silver in particular) is of great importance, as presented in the multiple available examples regarding the silver nanoparticle phytosynthesis. This also included the application of viticultural wastes [41,42] for developing antimicrobial solutions. Secondly, the scarce existing data were considered for the application of such nanoparticles for controlling vineyard pathogens [43]. Finally, the fact that ferns also represent a promising material for nanotechnology (namely for developing phytosynthesized nanoparticles) [44] was taken into consideration. According to our previous studies, the presence of phytosynthesized nanoparticles could lead to an enhancement of the antimicrobial properties of the extracts (when existent). This has been demonstrated by our group from experiments on silver nanoparticles phytosynthesized using *Raphanus sativus* L. extracts [45]. The presence of phytosynthesized nanoparticles also led to superior results as compared to the nanoparticles obtained using other green methods, such as radiation-assisted synthesis [46]. This is due, on the one hand, to the contribution of the secondary metabolites used for the reduction, and on the other to the capping of metallic nanoparticles [47]. This latter can have a higher influence on the final antimicrobial properties than the dimensions of the nanoparticles [46].

The objective of the paper is to evaluate the influence of two “green” recipes (an ethanolic extract of *D. filix-mas* and the dispersion of phytosynthesized nanoparticles obtained using the extract) on *U. necator* and *P. viticola* pathogens (regarding the frequency, intensity, and the attack degree), on the pith/wood rapport and on the soluble sugars, starch, and total water content, as well as on the length of the young shoots and number of grapes after the applied treatment. 

## 2. Materials and Methods

### 2.1. Collection and Treatment of Vegetal Material

The vegetal material was collected in 2018 from Valea Vâlsanului, Romania (45°20′41.1″ N; 24°44′06.0″ E, altitude of 735 m), being identified by Liliana Cristina Soare. A voucher specimen of the plant was preserved at the Argeş County Museum (voucher specimen no. 11330). The harvested plants were cleaned of impurities, washed with tap water, and dried at room temperature on paper sheets until a constant mass was achieved. The dried vegetal material was ground to a powder and preserved for further use [48].

### 2.2. Preparation and Characterization of Vegetal Extract (V1) and Silver Nanoparticles Solution (V2)

#### 2.2.1. Preparation of Sample V1 (Vegetal Extract) and V2 (Photosynthesized Nanoparticles)

The vegetal extracts (further encoded as V1) were obtained using temperature extraction by scaling up a recipe, which has been proved effective at the laboratory scale, for the case of 50 L reactors (Appendix A). The studied experimental conditions are as follows: 50% ethanol (Chimreactiv, Bucharest, Romania) as extraction solvent, ratio of vegetal material to solvent of 1:10 (*w*/*v*), extraction temperature of 50 °C, extraction time of 3 h, and continuous stirring. The water used for all experiments was distilled water, obtained in our laboratory.

After the extraction procedure, the extracts were allowed to cool down and filtered.

The phytosynthesis procedure was performed by direct mixing of a part of the extract with an equal quantity of 1 mM AgNO_3_ solution (Chimreactiv, Bucharest, Romania), thus obtaining the solution containing phytosynthesized silver nanoparticles, further encoded as V2. The procedure was performed under ambient conditions, in the presence of natural light, without stirring, the reaction being considered completed after 24 h.

#### 2.2.2. Characterization of Samples V1 and V2A

A small quantity of the two types of materials was prelevated for controlling the extract composition and the success of the phytosynthesis procedure, respectively. Both types of materials (the vegetal extract and the phytosynthesized nanoparticles solution) were stored in clean recipients at 4 °C for further use (Appendix A).

The control of the vegetal material was performed by determining the total phenolics content of the vegetal extract (the method being previously presented [45], with a brief description in the Appendix A) for the natural extract (further encoded as V1). The phytosynthesized nanoparticles (NPs) solution (further encoded as V2) was characterized using X-ray diffraction and transmission electron microscopy (TEM) (to define the obtained nanoparticles). XRD analyses were performed using a 9 kW Rigaku SmartLab diffractometer (Rigaku Corp., Tokyo, Japan, operated at 45 kV and 200 mA, Cu_Kα_ radiation—1.54059 Å), in scanning mode 2θ/θ, between 20 and 80° (2θ). The components were identified by comparison with ICDD data. The crystallite size was determined using the Scherrer equation:(1)$$Dp=\frac{\left(K\times \lambda \right)}{\left(\beta \times cos\theta \right)}$$
where Dp = the average size of the crystallites, K = the Scherrer constant (*K* = 0.94), β = width at half-height of the diffraction maximum, θ = Bragg angle, λ = the wavelength (1.54059Å).

Transmission electron microscopy analyses were performed using a Tecnai G2 F20 TWIN Cryo-TEM electron microscope (FEI Company, Hillsboro, OR, USA) at 300 kV acceleration voltage and 1Å resolution. For the analysis, a drop of nanoparticles-containing solution was directly placed on the copper grids, evaporated, and analyzed. The nanoparticle size distribution was examined via direct measurements of nanoparticles from TEM images (over 100 measurements) using ImageJ image analysis software (v. 1.53s, National Institutes of Health, Bethesda, MD, USA).

Other specific characteristics of the recipes constitute the subject of a pending patent application [49].

### 2.3. In Vitro Antimicrobial Studies

As a preliminary step, the two recipes were evaluated as antimicrobial agents using the agar well diffusion method [50,51] against *Plasmopara viticola*.

Briefly, diseased vine leaves were collected and washed with sterile water. The petiole was wrapped with absorbent cotton, and the small pieces of the abaxial surface of the leaves were placed in a Petri dish. Furtherly, they were cultured in a Laboshake shaker (Gerhardt GmbH & Co. KG, Königswinter, Germany) at 22 °C for 24–48 h, as previously described [7], and in concordance with literature data [52,53,54]. For the detached leaf assay, the sporangia were collected with a sterile brush. The suspension obtained was diluted into 10^5^ sporangia/mL. Potato Dextrose Agar plates (abbreviated PDA) from Sigma-Aldrich (Sigma Aldrich, St. Louis, MO, USA; composition: agar, 15 g/L, dextrose, 20 g/L and potato extract, 4 g/L supplemented with 0.1 g/L chloramphenicol, used to avoid the bacterial contamination) were inoculated with 1 mL inoculum of the *P. viticola*.

Wells in the PDA plates were made using a sterile borer at the size of 6 mm diameter, and then 50 μL of tested sample were placed in them. The Petri plates were incubated at 22 °C for 48 h in a shaker. As a negative control, ethanol: water = 1:1 (*v*/*v*) was used, and Pergado F 45 WG (Syngenta AG, Basel, Switzerland) was selected as a positive control, at 0.25% concentration (as recommended by the producer). Each experiment was carried out in triplicate.

### 2.4. Field Experiments

#### 2.4.1. Clones Used for Field Experiments

The field experiment was conducted between 2019–2021 on *V. vinifera* L. –Feteasca alba Feteasca alba 97 St., Feteasca neagra Feteasca neagra 6 St., Feteasca regala Feteasca regala 72 St., and Cabernet Sauvignon 131 St. Clones were located in vegetation pots in a greenhouse (Appendix A). The plants were own-rooted and intended for obtaining the viticultural propagating material from higher biological categories (the initial propagating material). For this experiment 120 plants were used, divided into 10 plants/experimental group for each clone. They were separated from each other by a paravane in order to apply differentiated treatments.

#### 2.4.2. Applied Treatments

In 2019, the control plants were treated with commercial pesticides Dithane M-45 (Indofil Industries Ltd., Mumbai, India), Flint max 75WG (Bayer AG, Leverkusen, Germany), and Sublic (Microspore Hellas, Athens, Greece) at the concentrations indicated by the producers (Table 1). The first treatment (19 June 2019—19/06/2019) was preventively applied and the other two treatments (10 August 2019—10/08/2019 and 21 August 2019—21/08/2019) were curative. In 2020, five treatments were applied. At the control plants, one-time treatments were applied with Dithane M-45 on 7 July 2020—07/07/2020, Thiovit Jet 80WG (Syngenta AG, Basel, Switzerland) on 16 July 2020—16/07/2020, and three treatments with Microthiol Special (Cerexagri—United Phosphorus Ltd., Vondelingenplaat, The Netherlands) fungicide (20 August 2020—20/08/2020, 27 August 2020—27/08/2020, 4 September 2020—04/09/2020) (Table 1). All the fungicide concentrations were calculated for one hectare, with dissolution in 800 L of water. For the 2019 and 2020 studies, both the alcoholic extract of *D. filix-mas* (V1) and the nanoparticles solution were tested and phytosynthesized using the alcoholic extract (V2); the extracts were applied undiluted, the solutions having a neutral pH (tested with pH paper). All treatments were performed by foliar application.

#### 2.4.3. Evaluation of Pathogen Symptoms

During the study, the appearance of the symptoms produced by the pathogens *P. viticola* and *U. necator* was monitored. During both years, only powdery mildew was formed, with the attack monitoring frequency (F), the intensity (I), and the attack degree (AD) being determined. These determinations were performed both for the control and for the experimental groups before and after the treatments’ application.

The frequency value was obtained by direct observations on the number of plants and leaves, and the collected data were computed using the following formula:
(2)$$F=\frac{n}{N}\times 100$$
where *n* is the number of plants or organs of the plant attacked by pathogens and *N* is the number of plants or organs observed.

The intensity of the attack was determined by reporting the leaf attacked area to the total area observed, thus being calculated by the degree and extend of the attack. To reproduce the intensity of the powdery mildew and downy mildew attack, the following classification scale was used, which included the attacked surface (%) and the severity (note) of the attack intensity: for 1–3% = 1; 4–10% = 2; 11–25% = 3; 25–50% = 4; 26–50% = 4; 51–75% = 5; 76–100% = 6.

To determine the coverage attack degree, the following formula was used:
(3)$$I=\sum^{\;}\frac{i\times f}{n}$$
where: *I* = the attacked note or surface (%); *f* = the number of cases with attack on each note; *n* = total number of cases with attack.

The determination of the *F*, *I*, and *AD* has a special importance in assessing the damage caused, in establishing the rhythm of treatments application, and in establishing the effectiveness of various methods and means of crop protection against pathogens.

The relation linking the intensity of powdery mildew to the total number of plants at which the observations were made was used in order to determine the attack degree. Its value expression is given by the relation:
(4)$$AD=\frac{F}{I}\times 100$$

#### 2.4.4. Biochemical Analyses and Grapevine Characteristics

At the same time, the evaluation of woody biological material of all considered clones was studied by determining the pith/wood (P/W) ratio, the total sugars (soluble sugars and starch), and the total water content. For the canes harvested from each studied grapevine plant, the dimensions of the pith and wood were measured at the base, middle, and top of the cane.

The assessment of the ratio between the pith and the wood was made considering the relationship presented in literature data [55]: P/W > 1− incompletely matured wood; P/W ≈ 1 – partially matured wood; P/W < 1 – wood well matured.

The canes’ water content (%) was obtained by drying the samples at 60–62 °C in the oven until constant mass [56].

The extraction of soluble sugars was carried out with warm 80% ethylic alcohol. The starch was extracted with 52% perchloric acid from the sediment resulting from the extraction of soluble sugars. A dosing of sugar was done with 0.2% anthrone prepared in 95% sulfuric acid. The absorbance reading of the samples was performed using a spectrophotometer at a wavelength of 620 nm [57]. The soluble sugars and starch contents were calculated in a percentage related to the dry material.

In order to provide a full image of the treatment influence on the grapevine productivity, the length of the young shoots and the number of grapes per plant were determined over the year 2020. The measurement of the young shoots was performed using a roulette, with the results being expressed in cm. The number of grapes was determined by direct counting.

The statistical significance was analyzed with IBM SPSS Statistics version 25 for Windows, one-way ANOVA. Different superscript letters mean statistically differences between the results obtained for different experimental groups at *p* < 0.05. The results are presented as means ± SD of three experiments. For graphical representations, the OriginPro 2018 Data Analysis and Graphing Software (OriginLab Corporation, Northampton, MA, USA) was used.

## 3. Results and Discussions

### 3.1. Characterization of Extract and Extract Containing AgNPs

The control of the natural recipes proposed for treatment was performed, as presented in the previous section, using a phytochemical assay (for the natural extract). The evaluation of the nanoparticles crystalline structure was performed using X-ray diffraction. Their morphological properties were evaluated using transmission electron microscopy. The total phenolics content determined for sample V1 was 36 ± 1.25 mg/L GAE (gallic acid equivalents).

The silver in V2 was evaluated by XRD analysis. Approximately 10 mL of V2 were centrifuged and the obtained precipitate was deposited on specific glass support for analysis. Figure 1 presents the diffractogram of the analyzed sample. 

The present phases identification was based on the comparison with ICDD entries. This resulted in confirmation of the silver nanoparticles synthesis (PDF card no. 01-087-0720) by the apparition of diffraction peaks corresponding to the (111), (200), (220), and (311) diffraction planes. Additionally, the presence of a secondary silver oxide phase was confirmed by the apparition of silver oxide diffraction peaks, corresponding to the diffraction planes (110), (111), (220), and (221) (PDF card no. 01-078-5867), with peaks marked by an asterisk in Figure 1. This phase is due to the silver oxidation occurring during the pre-treatment stage of the analysis. The crystallite size determined using Equation (1) applied to the parameters of the (111) diffraction plane was 6.72 nm.

The electron microscopy evaluation (Figure 2) revealed the formation of spherical and quasi-spherical nanoparticles with an average diameter of 17 nm, although larger particles and clusters of nanoparticles can also be observed. 

The obtained results support the formation of silver nanoparticles using fern extract. When comparing the results obtained with previous works, it can be noticed that, using the presented proposed method, larger NPs are obtained as compared to the ones phytosynthesized using classical temperature or microwave extracts of *Asplenium scolopendrium* L. leaves (under 14 nm) [46]. However, when comparing the results with the ones obtained using other types of vegetal materials, smaller dimensions were obtained in the present study when compared with *Aconitum toxicum* Reichenb rhizomes (over 50 nm) [58] and in the same range with those obtained using *Raphanus sativus* L. leaves [45].

### 3.2. In Vitro Antimicrobial Evaluation

Our previous works in the area of phytosynthesized nanoparticles [45,46,58] suggested that the silver nanoparticles obtained using ferns or other green leaves exhibit a potent antimicrobial activity, even against crops pathogens (such as *Venturia inaequalis* and *Podosphaera leucotricha*) [45]. In order to test this hypothesis, an in vitro evaluation of the antimicrobial potential of the two solutions (V1 and V2) was performed before field experiments on downy mildew (*Plasmopara viticola*). The obtained results are presented in Figure 3.

The laboratory experiments suggest that the fern extract is not active against downy mildew (statistically similar results being obtained for V1 and negative control). A significant antimicrobial effect was observed for the V2 sample, for which an inhibition zone of approximately half of the positive control was observed. 

Similar observations were made by Zheng et al. [59] (when applying Ag-SiO_2_ core-shell nanoparticles against several phytopathogenic fungi) and by Wang et al. [60] (using a graphene oxide-Fe_3_O_4_ nanocomposites). In the latter case, the effect was assigned by the authors to the blocking of sporangia’s water channels through the surface adsorption of the nanocomposite [60,61]. Rashad et al. [62] used silica nanoparticles to control downy mildew and assigning the protective effect to the influence of the NPs on stomatal closure, stomatal area, and stomatal pore area.

### 3.3. Results of Field Experiments

The experimental observations on the occurrence of powdery mildew and downy mildew in the studied grapevine plants, together with the time and space disease evolution, were monitored for highlighting different levels of attack for each clone. On the studied grapevines, only powdery mildew appeared in the studies from 2019 and 2020. Thus, on 9 August, 19 August, and 26 August 2019, the F, I, and AD evaluations were performed for estimating powdery mildew on the grapevine plants studied (Figure 4). The symptoms were different from one genotype to another. For Feteasca alba Feteasca alba 97 St., a powdery mildew attack in all three groups (control and experimental) was registered (on 9 August 2019). Furthermore, after the first treatment application, considering the symptoms’ appearance, the attack was stopped only in the case of V1 (when the plants were sprayed with alcoholic extract of *D. filix-mas*). After the third treatment, the attack was stopped in all three groups (control and experimental). The control plants of Feteasca neagra Feteasca neagra 6 St. did not register a powdery mildew attack. On the other hand, in the case of plants from V1 and V2 experimental groups, the attack of powdery mildew stopped after the last treatment application.

Regarding the Feteasca regala Feteasca regala 72 St. clone, the plants of all three groups were affected by powdery mildew. After the second treatment, only those from the V1 experimental group were still affected. However, in their case, the attack of powdery mildew was stopped after the third treatment application.

In the case of the Cabernet Sauvignon 131 St., only the plants from control and V2 experimental groups were affected by powdery mildew and the attack was stopped after the second treatment. The attack of powdery mildew appeared after the second treatment only on new vegetative growths (tops and lateral shoots). Therefore, 5 days after the application of the last treatment (26 August 2019), the attack of powdery mildew was stopped on all grapevine plants (control, V1, V2).

The same observations regarding the appearance of the main pathogens were made in 2020. The symptoms were visible for the first time at the beginning of July on Feteasca neagra Feteasca neagra 6 St. for the experimental groups V1 and V2 due to higher plant growth (thus creating a favorable climate for the development of powdery mildew) as compared to the control (where the vegetative growths were reduced). After the application of 5 treatments, the attack was stopped for the plants in experimental groups V1 and V2, while for the control group the attack was stopped after the third treatment. The five treatments application led to the protection of Feteasca regala Feteasca regala 72 St. plants from V1 and V2 groups against the attack of powdery mildew, compared to the control where the attack was stopped after the application of the last chemical treatment. Following the application of treatments on Feteasca alba Feteasca alba 97 St., it was found that throughout the vegetation period of the plants belonging to the V2 experimental group, no disease-specific symptoms were formed. Furthermore, in the other two groups the degree of attack was similar, with 0.23% for control and 0.26% for V1. After the third treatment, the attack of powdery mildew stopped in both groups. At Cabernet Sauvignon 131 St., no downy mildew and powdery mildew attack was recorded during the entire vegetation period, regardless of the experimental group. Therefore, the two experimental groups (V1 and V2) ensured a very good phytosanitary condition, identical to the chemical treatments (control) (Figure 5).

Similar results were obtained by other authors on Thompson Seedless grapevine, a significant reduction in the abnormal alterations induced by the downy mildew on grapevine leaves upon SiNPs spraying being recorded [62]. Additionally, nanoparticles of silicon and titanium were used to reduce powdery mildew infection and development at wheat plants [63], while silver nanoparticles were used for *Phytophthora parasitica* control [64]. For decades, pesticides have been an essential part of agriculture [65]. However, when applying a phytosanitary treatment, 30% of the applied substance is scattered in the atmosphere and about 30% falls on the ground, with these fractions contributing to air and soil pollution [66]. 

The attack degree of the *U. necator* pathogen was different from one genotype to another. Furthermore, during the studied years 2019 and 2020, it has been observed after the last treatment that the powdery mildew attack stopped on all grapevine plants, both in the control and in the case of the experimental group treated with V1 and V2 solutions. No downy mildew attack was recorded during the entire vegetation period, regardless of the experimental group.

Analyzing the wood from a morphological point of view, it was found that in 2019 and 2020 all the canes were well matured at all experimental groups. However, determinations at the base, middle, and top level of the cane for all the studied grapevine clones were performed. The purpose was to find out if the analyzed canes recorded differences in the P/W rapport in the same experimental groups along their length. 

The study of the influence of biological products’ application against downy and powdery mildew on the processes of cane’s maturation is of high practical importance. This is due to the fact that the grapevine plants studied in this paper are used as initial propagating material. In the Feteasca regala Feteasca regala 72 St. and Cabernet Sauvignon 131 St. genotypes, the measurements on the pith/wood ratio did not reveal significant variations between the treatments with V1 and V2 solutions, as compared to the chemical treatment (control), during the two years of study on the entire length of the canes. In Feteasca alba Feteasca alba 97 St., for the V1 experimental group, the ratio increased significantly in the upper part of the canes after the first year of treatment only. However, the treatment with the V2 solution led to significant increasing or decreasing variations along the entire length of the cane in 2020 and only at the upper part in 2021. The Feteasca neagra 6 St. genotype behaved almost similarly (Table 2).

In the case of the Feteasca alba 97 St. clone (in 2020), the control recorded a decrease in the diameter of the top of the cane by 28.71% as compared to its base. However, for the plants treated with the V1 solution, the decrease was only 0.79%, and for the plants treated with the V2 solution there was an increase in its diameter by 41%. In the same year, Feteasca neagra 6 St. registered an increase of 8.65% in the case of the control, 20.67% for the plants treated with V1, and a slight decrease of 0.78% for plants treated with V2. The control of the Feteasca regala 72 St. registered an increase of 24.03%, while for the plants treated with V1 and V2, the increases were 10.26% and 19.74%. The fourth clone, Cabernet Sauvignon 131 St., registered an increase of 34% in the control, 28.01% for the treatment with V1, and a decrease of 7.35% for the treatment with V2.

The pith/wood ratio had subunit values both in the case of chemical fungicide treatments (control) and in the experimental groups V1 and V2. Thus, by maintaining the health of plants, the normal physiological course of vegetative phenophases was favored. Similar results were obtained in other studies regarding the use of biostimulants for an increased tolerance to the virus diseases in grapevine [67].

Accumulation of the reserve substances in the canes wooden tissue is very important. Its maturation corresponds to the period when the total carbohydrates (soluble sugars and starch) reach the maximum. This period corresponds to the grapevine dormancy [56]. It varies with the variety vigor, the load of the shoots, and the climatic conditions. The maturation of the canes is a physiological process of reserve substances accumulation in the shoots (the maximum starch content when the leaves fall is about 11%). Meanwhile, the water content decreases from 80% to 45–50%, a level that ensures the viability of tissues until the spring of the next year [68].

In order to find out whether the treatments applied in 2019 and 2020 influenced the degree of wood maturation, biochemical determinations (total sugars, water and dry matter) were performed in February 2020 and February 2021. In the second year of the study, the biochemical analyses of the carbohydrate content (soluble sugars and starch), as well as the total water content of the canes collected from the studied grapevine genotypes, showed only significant sporadic differences in the experimental groups as compared to the control group. The treatment with nanoparticles solution (V2) led to a significant increase of the starch content in Feteasca alba 97 St., and an increase of soluble and total carbohydrates in Cabernet Sauvignon 131 St. An increase of the total water content in Feteasca neagra 6 St. was recorded for plants treated with both V1 and V2 solutions (Table 3).

The experimental group V2 increased the frost resistance of the Cabernet Sauvignon 131 St. clone by the starch content increasement. Additionally, an increase in the soluble sugars was observed for the same clone (only after two years of treatment—2021). This variation could be explained by the NPs accumulation over the two treatment years and their involvement in the photosynthesis and sugar transformation processes. This has also been observed by other authors for the case of nanoparticles treatments [69,70,71,72]. In the canes, the conversion of starch into soluble carbohydrates is faster at resistant varieties. However, in the sensible ones, the starch content is kept at a higher level. Feteasca neagra, Feteasca regala, Cabernet Sauvignon, and Feteasca alba genotypes are considered resistant to frost [73]. Ruano-Rosa et al. [43] also used silver nanoparticles for pathogen control (against powdery mildew on the grapevine). As compared to the results obtained by them, the results presented in this manuscript are more promising. The starch content fluctuates in the roots and trunks during the vegetation period, with the lowest value being recorded between budbreak and flowering and the highest values between harvest and leaf fall [74].

The maturation of the grapevine canes is of major importance, being related to the frost resistance and to the quality of the scion and rootstock for the planting material production. The first studies on the cane maturation process are attributed to Gouin and Andouard. They studied the dry matter accumulation dynamics (in the year 1899) and nitrogen and starch content in canes (in the year 1901) [75].

Regarding the carbohydrates’ dynamics during the dormancy, starting from December, the starch stored as a reserve substance was gradually hydrolyzed, reaching minimum values in February, while the reducing carbohydrates increase. According to several authors, when considering the starch cumulated with soluble sugars, it is estimated as a minimum level of 12% [76]. This provides nutritional support for starting a new vegetative cycle. In our study, the level of carbohydrates and total water includes the canes collected from plants subjected to treatments against downy and powdery mildew at an optimal level of maturation. This level was suitable for use in the propagation process, with total carbohydrates registering values between 16% and 20%. The same genotypes represented by grapevines stored under the same conditions (greenhouse) as those used in this study behaved similarly, with total carbohydrate values between 14 and 16%, depending on the climatic and cultivation conditions of that year [77].

In conclusion, the subunit ratio, the water content of over 46%, and the soluble carbohydrates of over 16% put the canes collected from the studied plants at an optimal level of maturation that makes them suitable in the multiplication process.

In order to establish the effect of the performed treatment on the productivity of the grapevine, the determination of young shoots length and grape number were performed over the year 2020 (Figure 6).

During 2020, the growth of young shoots and the number of grapes were different from one genotype to another. Among the studied genotypes, the most significant increase in the shoot growth was recorded for the Feteasca neagra 6 St. treated with V2 (for which the lowest number of grapes among the treated genotypes was also recorded, although higher than the control). The experimental groups V1 and V2 registered a higher number of grapes compared to the control in all studied genotypes; for the genotypes F. alba 97 St and F. neagra 6 St., a higher quantity of grapes upon treatment with V2 was recorded as compared to V1, while for F. regala 72 St. and C. Sauvignon 131 St., treatment with V1 led to higher amounts of grapes compared to V2 (Figure 6). As such, the performed treatments could have a biostimulant effect on some of the studied genotypes, an aspect which needs to be further studied. According to the available literature data, the nanoparticles have different pathogen inhibition pathways depending on the nature of the nanomaterial. For example, silica NPs were found to enhance the resistance of the plants to the action of the pathogens. More precisely, their application led to an increase in photosynthetic pigments and overexpression of plant hormones regulating the accumulation of relevant secondary metabolites. Similar observations were made by Farhat et al. [63] for SiNPs, while the action of the TiO_2_ NPs was assigned by the authors to their photocatalytic properties. For AgNPs, literature data [64] suggest a complex mechanism, in which the inhibition of respiratory chain enzymes, the disruption of ribosome biogenesis, and the impairment of genes involved in oxidative stress all coexist.

In the case of the present study, the most probable explanation is a mixture of the two types of action. Thus, on the one hand, the silver nanoparticles are known to exhibit a potent antimicrobial effect against different types of microorganisms. These mechanisms have been previously presented by our group [47] and were found to be mainly based on disruption of cellular membrane functionality and generation of reactive oxygen species. These properties are well-connected to the properties of the NPs (size, morphology, etc.) and to the synthesis method, as the secondary metabolites acting as capping agents also enhance their properties. On the other hand, the applied treatments (either as fern extract or as silver nanoparticles phytosynthesized using the fern extract) led to an increase in several biochemical parameters (starch, sugars, and water content), as well as in productivity (grape number) and vigor parameters (length of young shoots) in a clone-dependent manner. These results would suggest a biostimulant effect which could contribute to the increase in plant resistance to pathogens, an aspect which needs further investigation.

## 4. Conclusions and Future Perspectives

The present study aimed to evaluate the protective effect on grapevine of a fern extract and of silver phytosynthesized using the same extract. Both types of materials revealed a significant decrease (by comparison with the chemical pesticide used in the experiments) of the frequency, intensity, and attack degree of the pathogen *Uncinula necator* (responsible for the apparition of powdery mildew). This observation was suggested by the laboratory results against *Plasmopara viticola* (pathogen responsible for the apparition of downy mildew).

The determinations regarding the pith/wood ratio of the canes showed that all the plants studied have developed very well-matured canes in all treatment options. Statistically significant differences were recorded for the Feteasca alba 97 St. clone treated with the nanoparticle solution in 2020 (lower for base and middle, higher for top level of cane, as compared to the control), while marginal differences were also recorded for 2021 (higher for all determination areas). Similar observations were made for the Feteasca neagra 6 St. clone (except for the year 2021, top zone, where significant lower values were recorded). For Feteasca regala 72 St., significantly higher values for middle and top 2021 were recorded, while for Cabernet Sauvignon 131 St., significantly lower values for middle 2020 were recorded. For all the clones, the rest of the determinations had marginal variations. 

Regarding the biochemical determinations, significant variations were observed for soluble sugars (increase for Feteasca neagra and Cabernet Sauvignon/second year), for starch (increase for Feteasca alba and Cabernet Sauvignon in 2021 for both clones), for total sugars (increase for Feteasca alba and Feteasca neagra in 2021 for both clones), and for total water content (increase for Feteasca alba and Feteasca neagra in 2021 for both clones), respectively. More than that, the biostimulant effect of the fern extract and of the silver nanoparticles solution was further confirmed by the determinations on the number of grapes produced by each clone and on the length of the young shoots. 

Based on the findings of this study, in which positive results were obtained using hydroalcoholic extract of *D. filix-mas* and phytosynthesized nanoparticles mixture as preventive and curative treatments grapevine against downy mildew and powdery mildew in protected area, it is recommended to extend the research in vineyards. Further studies are also necessary in order to clarify the exact mechanisms involved in the clone-dependent protective potential of the proposed solutions.

## Figures and Tables

**Figure 1 materials-15-08188-f001:**
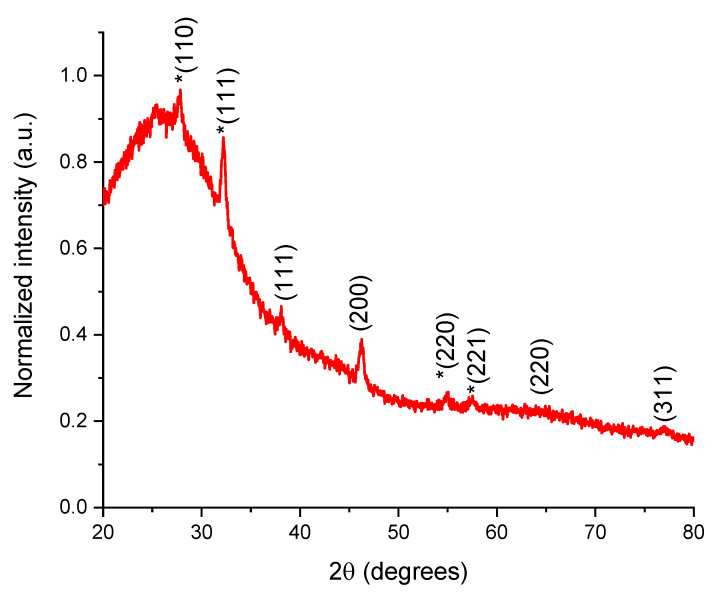
X-ray diffractogram of the dispersed components in the hydroalcoholic extract of D. *filix-mas* containing phytosynthesized silver nanoparticles. Diffraction planes of Ag^0^ and Ag_2_O (silver oxide planes marked by an asterisk) with Miller indexes *(hkl)* are presented on the diffractogram.

**Figure 2 materials-15-08188-f002:**
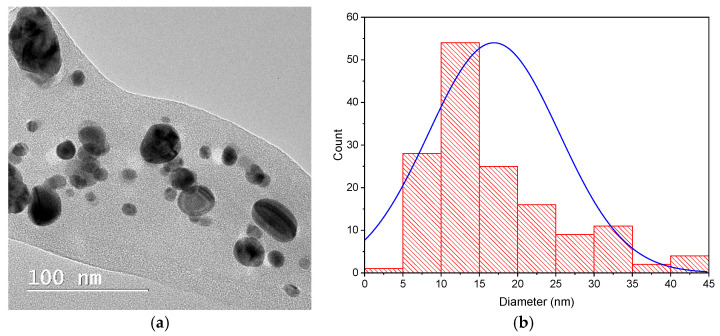
TEM image (**a**) and size distribution (**b**) of the dispersed components in the hydroalcoholic extract of D. *filix-mas* containing phytosynthesized silver nanoparticles.

**Figure 3 materials-15-08188-f003:**
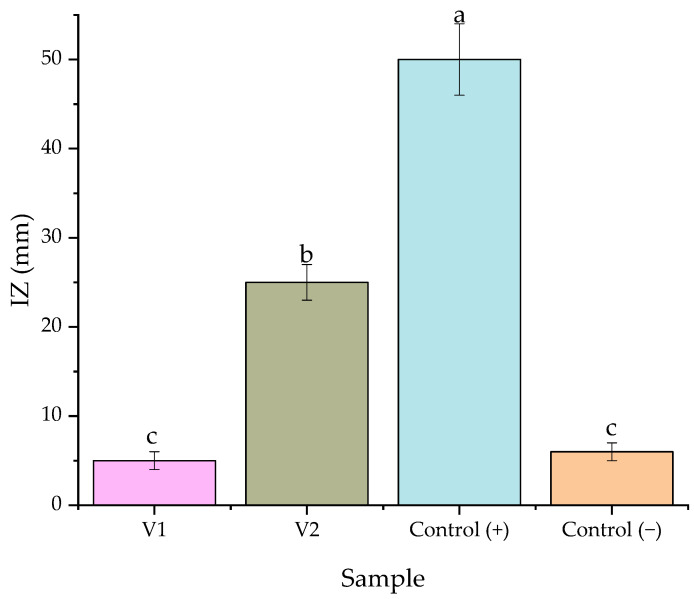
Antimicrobial potential of V1 and V2 samples against *P. viticola* in laboratory conditions. Control (+)—positive control (Pergado F 45 WG), Control (−)—negative control (hydroalcoholic solution). The values represent means ± SD. Means of the same parameter without a common lowercase letter (a to c) differ (*p* < 0.05) as analyzed by one-way ANOVA and the TUKEY test.

**Figure 4 materials-15-08188-f004:**
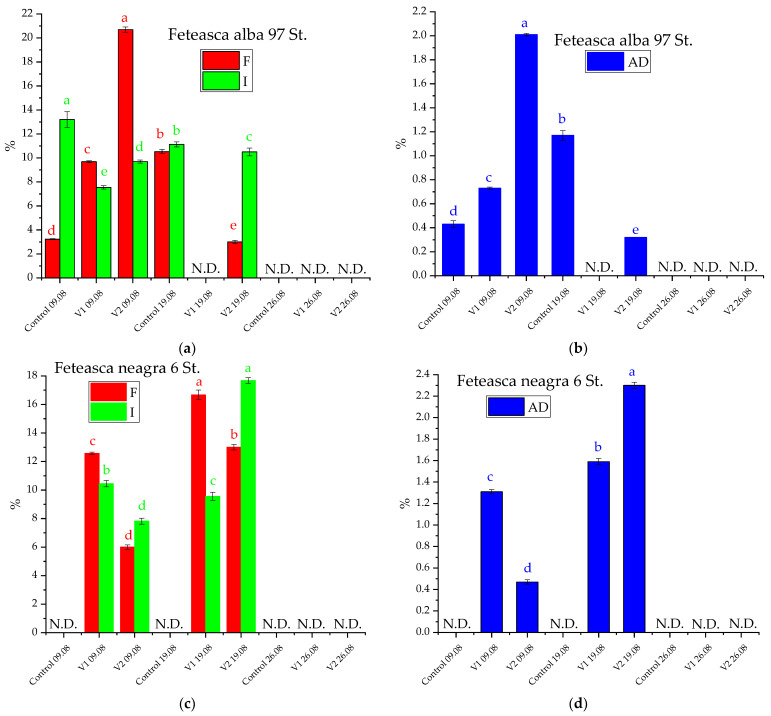
Frequency (F), intensity (I), and attack degree (AD) evolution of *U. necator* during successive treatments in 2019 on the studied clones: (**a**,**b**): Feteasca alba 97 St; (**c**,**d**): Feteasca neagra 6 St; (**e**,**f**): Feteasca regala 72 St.; (**g**,**h**): Cabernet sauvignon 131 St.. The values represent means ± SD. Means of the same parameter (same color) without a common lowercase letter (a to e) differ (*p* < 0.05) as analyzed by one-way ANOVA and the TUKEY test; V1 = alcoholic extract of *D. filix-mas*, V2 = alcoholic extract of *D. filix-mas* with silver nanoparticles, C—control (chemical pesticide); N.D.—not detected (no pathogen attack registered).

**Figure 5 materials-15-08188-f005:**
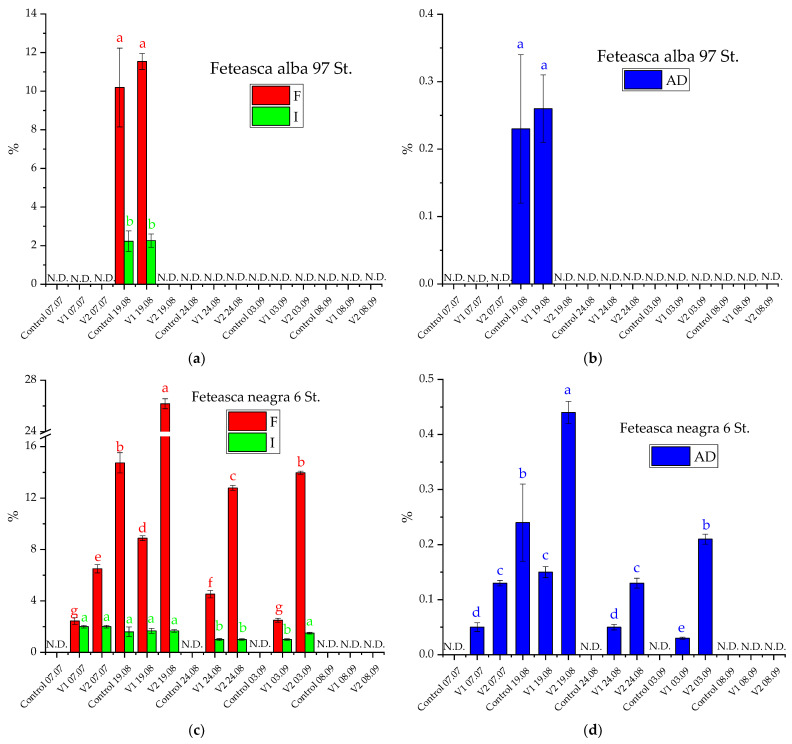
Frequency (F), intensity (I), and attack degree (AD) evolution of *U. necator* during successive treatments in 2020 on the studied clones: (**a**,**b**): Feteasca alba 97 St; (**c**,**d**): Feteasca neagra 6 St; (**e**,**f**): Feteasca regala 72 St. The values represent means ± SD. Means of the same parameter (same color) without a common lowercase letter (a to g) differ (*p* < 0.05) as analyzed by one-way ANOVA and the TUKEY test; V1 = alcoholic extract of *D. filix-mas*, V2 = alcoholic extract of *D. filix-mas* with silver nanoparticles, C—control (chemical pesticide); for the Cabernet Sauvignon 131 St. clone, all the registered results were null; N.D.—not detected (no pathogen attack registered).

**Figure 6 materials-15-08188-f006:**
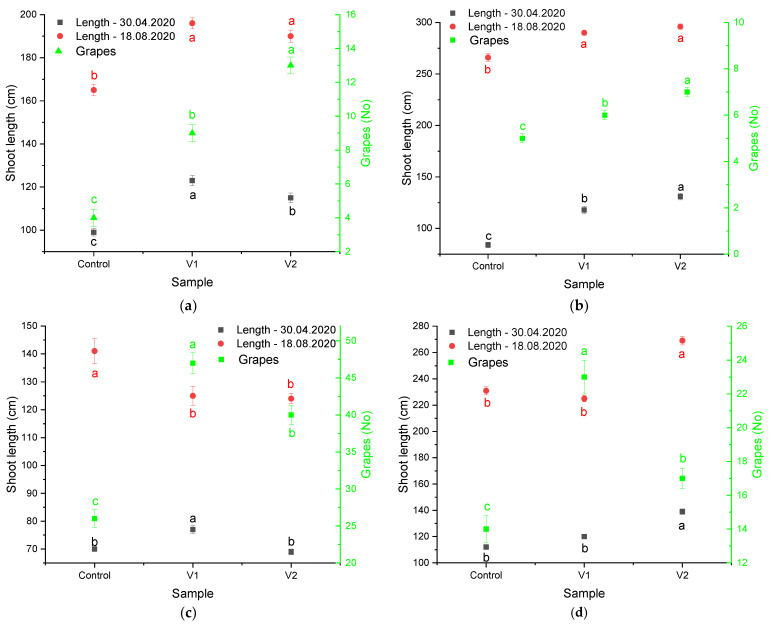
Influence of the applied treatments on young shoot length and number of grapes on the studied clones: (**a**) Feteasca alba 97 St; (**b**) Feteasca neagra 6 St; (**c**) Feteasca regala 72 St.; (**d**). Cabernet Sauvignon 131 St. The values represent means ± SD. Means of the same parameter (same color) without a common lowercase letter (a–c) differ (*p* < 0.05) as analyzed by one-way ANOVA and the TUKEY test; V1 = alcoholic extract of *D. filix-mas*, V2 = silver nanoparticles phytosynthesized using the alcoholic extract of *D. filix-mas*, Control—chemical pesticide.

**Table 1 materials-15-08188-t001:** Treatments application to the experimental variants.

Year	Treatment Time	Date	Treatment Type/ Diseases	Control Treatment	Examined Treatments	
2019	1st	19/06	Preventive/ downy mildew	Dithane 0.2% (80% mancozeb)	V1	V2
	2nd	10/08	Curative/ powdery mildew	Flint max 75 WG (500 g/kg Tebuconazol + 250 g/kg Trifloxistrobin), 0.16 kg/ha	V1	V2
	3rd	21/08	Curative/ powdery mildew	Sublic (*Bacillus* sp.) 1.2–1.5 L/ha + NutryAction 100–200 mL/hL (brown algae, Microspore Hellas, Athens, Greece)	V1	V2
2020	1st	07/07	Curative/powdery mildew	Thiovit Jet (80% sulfur), 3 kg/ha	V1	V2
	2nd	16/07	Curative/powdery mildew	Thiovit Jet (80% sulfur), 3 kg/ha	V1	V2
	3rd	20/08	Curative/ powdery mildew	Microthiol Special (800 g/kg sulfur), 20–30 g/10 L water	V1	V2
	4th	27/08	Curative/ powdery mildew	Microthiol Special (800 g/kg sulfur), 20–30 g/10 L water	V1	V2
	5th	04/09	Curative/ powdery mildew	Microthiol Special (800 g/kg sulfur), 20–30 g/10 L water	V1	V2

**Table 2 materials-15-08188-t002:** The pith/wood ratio in the three zones of the cane (base, middle, top), at Feteasca alba 97 St., Feteasca neagra 6 St., Feteasca regala 72 St., Cabernet Sauvignon 131 St. clones. V1 = alcoholic extract of *D. filix-mas*, V2 = silver nanoparticles phytosynthesized using *D. filix-mas* extract, C—control (chemical pesticide). The measurements were performed in February 2020 and February 2021 ^1^.

Group/Zone	2020			2021		
	Control	V1	V2	Control	V1	V2
Feteasca alba 97 St.						
Base	0.53 ± 0.09 ^a^	0.51 ± 0.03 ^ab^	0.38 ± 0.03 ^c^	0.48 ± 0.07 ^b^	0.49 ± 0.05 ^ab^	0.51 ± 0.01 ^ab^
Middle	0.56 ± 0.04 ^a^	0.57 ± 0.10 ^a^	0.41 ± 0.05 ^b^	0.39 ± 0.11 ^b^	0.44 ± 0.11 ^b^	0.46 ± 0.04 ^b^
Top	0.38 ± 0.07 ^c^	0.50 ± 0.08 ^ab^	0.54 ± 0.04 ^a^	0.38 ± 0.03 ^c^	0.38 ± 0.14 ^c^	0.44 ± 0.09 ^bc^
Feteasca neagra 6 St.						
Base	0.39 ± 0.09 ^bc^	0.45 ± 0.11 ^ab^	0.38 ± 0.08 ^bc^	0.37 ± 0.04 ^c^	0.47 ± 0.06 ^a^	0.47 ± 0.08 ^a^
Middle	0.49 ± 0.06 ^a^	0.46 ± 0.05 ^a^	0.38 ± 0.02 ^b^	0.36 ± 0.10 ^b^	0.34 ± 0.05 ^b^	0.32 ± 0.09 ^b^
Top	0.43 ± 0.02 ^b^	0.54 ± 0.06 ^a^	0.38 ± 0.15 ^b^	0.39 ± 0.04 ^b^	0.44 ± 0.13 ^b^	0.27 ± 0.03 ^c^
Feteasca regala 72 St.						
Base	0.39 ± 0.04 ^b^	0.39 ± 0.14 ^b^	0.39 ± 0.01 ^b^	0.49 ± 0.04 ^a^	0.53 ± 0.14 ^a^	0.55 ± 0.06 ^a^
Middle	0.43 ± 0.06 ^cd^	0.51 ± 0.06 ^b^	0.42 ± 0.07 ^d^	0.48 ± 0.02 ^bc^	0.51 ± 0.10 ^b^	0.57 ± 0.02 ^a^
Top	0.48 ± 0.04 ^a^	0.43 ± 0.08 ^a^	0.47 ± 0.05 ^a^	0.35 ± 0.09 ^b^	0.47 ± 0.11 ^a^	0.44 ± 0.05 ^a^
Cabernet Sauvignon 131 St.						
Base	0.27 ± 0.08 ^d^	0.36 ± 0.09 ^bc^	0.31 ± 0.02 ^cd^	0.37 ± 0.06 ^ab^	0.42 ± 0.02 ^a^	0.35 ± 0.01 ^bc^
Middle	0.34 ± 0.04 ^c^	0.44 ± 0.04 ^a^	0.30 ± 0.02 ^c^	0.39 ± 0.02 ^b^	0.39 ± 0.08 ^b^	0.41 ± 0.05 ^ab^
Top	0.37 ± 0.09 ^b^	0.46 ± 0.07 ^a^	0.29 ± 0.05 ^c^	0.43 ± 0.06 ^ab^	0.45 ± 0.08 ^a^	0.47 ± 0.06 ^a^

^1^ Values are means ± SEM, n = 3 per treatment group. Means in a row without a common superscript letter (a to d) differ (*p* < 0.05) as analyzed by one-way ANOVA and the TUKEY test.

**Table 3 materials-15-08188-t003:** Biochemical analyses at Feteasca alba 97 St., Feteasca neagra 6 St., Feteasca regala 72 St., Cabernet Sauvignon 131 St. plants. V1 = alcoholic extract of *D. filix-mas*, V2 = silver nanoparticles phytosynthesized using *D. filix-mas* extract, C—control (chemical pesticide). The measurements were performed in February 2020 and February 2021 ^1^.

Group/Parameter	2020			2021		
	Control	V1	V2	Control	V1	V2
Feteasca alba 97 St.						
Soluble sugars (%)	16.20 ± 3.54 ^a^	14.49 ± 0.39 ^bc^	14.53 ± 2.58 ^ac^	15.75 ± 0.53 ^ab^	14.19 ± 0.64 ^bc^	13.61 ± 0.50 ^c^
Starch (%)	3.30 ± 0.82 ^cd^	2.79 ± 1.21 ^d^	3.56 ± 0.95 ^c^	4.36 ± 0.39 ^b^	4.43 ± 0.42 ^b^	5.19 ± 0.62 ^a^
Total sugars (%)	19.50 ± 3.52 ^ab^	17.29 ± 2.14 ^b^	18.09 ± 2.47 ^ab^	20.11 ± 3.52 ^a^	18.62 ± 2.14 ^ab^	18.80 ± 2.47 ^ab^
Total water (%)	59.06 ± 3.89 ^a^	59.09 ± 3.33 ^a^	54.23 ± 13.18 ^ab^	50.38 ± 0.33 ^b^	52.02 ± 1.89 ^b^	52.69 ± 0.99 ^b^
Feteasca neagra 6 St.						
Soluble sugars (%)	17.46 ± 0.65 ^a^	16.27 ± 0.61 ^b^	16.66 ± 0.72 ^b^	15.28 ± 0.28 ^c^	14.61 ± 0.59 ^d^	16.12 ± 0.68 ^b^
Starch (%)	2.58 ± 1.70 ^c^	2.38 ± 0.91 ^c^	2.79 ± 0.64 ^bc^	3.78 ± 0.28 ^a^	3.12 ± 0.85 ^ac^	3.47 ± 0.56 ^ab^
Total sugars (%)	20.04 ± 2.34 ^a^	18.65 ± 1.39 ^bc^	19.45 ± 1.15 ^ab^	19.06 ± 0.15 ^ab^	17.73 ± 0.89 ^c^	19.59 ± 0.73 ^ab^
Total water (%)	56.16 ± 8.58 ^a^	54.91 ± 0.80 ^ab^	53.29 ± 7.86 ^ab^	46.50 ± 1.13 ^c^	51.78 ± 1.26 ^ab^	51.19 ± 3.19 ^b^
Feteasca regala 72 St.						
Soluble sugars (%)	15.11 ± 0.81 ^b^	15.63 ± 0.65 ^a^	15.23 ± 0.32 ^ab^	12.01 ± 0.65 ^d^	12.78 ± 0.41 ^c^	11.69 ± 0.17 ^d^
Starch (%)	3.17 ± 1.50 ^bc^	2.21 ± 1.41 ^c^	2.71 ± 0.66 ^c^	4.27 ± 1.11 ^ab^	4.83 ± 1.94 ^a^	4.95 ± 0.54 ^a^
Total sugars (%)	18.27 ± 0.11 ^a^	17.83 ± 0.47 ^ab^	17.95 ± 0.69 ^ab^	16.28 ± 0.59 ^c^	17.61 ± 0.72 ^b^	16.63 ± 0.76 ^c^
Total water (%)	55.42 ± 6.69 ^ab^	58.01 ± 0.96 ^a^	58.91 ± 6.13 ^a^	51.78 ± 2.52 ^bc^	52.39 ± 1.94 ^bc^	51.69 ± 2.38 ^c^
Cabernet Sauvignon 131 St.						
Soluble sugars (%)	14.03 ± 4.69 ^ab^	15.73 ± 4.48 ^a^	14.70 ± 1.48 ^ab^	10.42 ± 0.39 ^c^	10.93 ± 0.74 ^c^	12.68 ± 0.59 ^bc^
Starch (%)	2.87 ± 5.39 ^b^	3.11 ± 2.86 ^b^	3.75 ± 3.80 ^ab^	5.95 ± 0.43 ^a^	5.07 ± 0.61 ^ab^	6.15 ± 0.74 ^a^
Total sugars (%)	16.90 ± 0.51 ^b^	18.84 ± 0.47 ^a^	18.45 ± 0.54 ^a^	16.36 ± 0.51 ^c^	16.00 ± 0.47 ^c^	18.83 ± 0.54 ^a^
Total water (%)	55.85 ± 5.66 ^a^	54.87 ± 2.15 ^a^	54.75 ± 0.37 ^a^	48.82 ± 2.74 ^b^	49.37 ± 0.85 ^b^	48.18 ± 1.14 ^b^

^1^ Values are means ± SEM, n = 3 per treatment group. Means in a row without a common superscript letter (a to d) differ (*p* < 0.05) as analyzed by one-way ANOVA and the TUKEY test.

## Data Availability

The data presented in this study are available on request from the corresponding authors.

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
