# Peer review of "Grapevine Plants Management Using Natural Extracts and Phytosynthesized Silver Nanoparticles"

_materials, 2022, doi:10.3390/ma15228188_

Round 1
Reviewer 1 Report (Previous Reviewer 1)
The author revised the manuscript very well and addressed most of my comments.
Hence I agree with the acceptance of this paper.
However, there are still a lot of grammatical language errors that the author needs to check.
In addition to that, there are several peaks in XRD not discussed. The author should discuss all the peaks and relate it with other findings.
Author Response
"Please see the attachment."

Reviewer 2 Report (Previous Reviewer 2)
Please see the attached file.

Author Response
"Please see the attachment."

Reviewer 3 Report (Previous Reviewer 3)
This manuscript has been greatly improved by several rounds of revisions, and my concerns have been addressed. The current version of this manuscript is within the acceptable range for consideration.
Author Response
"Please see the attachment."

This manuscript is a resubmission of an earlier submission. The following is a list of the peer review reports and author responses from that submission.
Round 1
Reviewer 1 Report
The experiments seem conducted appropriately, but the presentation is seriously flawed. I accordingly could not review the scientific validity enough. I may reconsider after the styles of this paper are sufficiently improved.
1. Manuscript still has a lot of grammatical errors and missing sentence meanings.
2. The nanoparticle used is only silver nanoparticles. The nanoparticles should be specified to silver nanoparticle throughout the manuscript, especially the title and abstract.
3. Page 1, line 35. I recommend changing the sentence to describe the global importance of viticulture. This journal is not a domestic Romanian journal, and readers of other countries should have interest on the work described.
4. I suggest removing the last 3 lines 42-44 of the abstract (in conclusion -------”) to highlight the main points and results in the abstract.
5. In the introduction section last few lines 129-130 author mentioned According to our previous studies, the presence of photosynthesized nanoparticles ………., Author should remove these lines from this part and add them to the previous paragraph with a clear explanation of the previous work and difference between their previous work and the current study.
6. Experimental section should be divided into appropriate sections, materials, measurements, and each method. I could not understand the contents enough, because various matters are included without sections.
7. At least, the following matters must be indicated. Most of the descriptions are insufficient.
8. The detailed methods for the synthesis of silver nanoparticles and the preparation of V1 (how the extraction was conducted) and V2
9. The content of silver. What ratio? Weight, mol? I could not understand the ratio of the extract (solid content? Including liquid?) and precursor (what was used?).
10. The manufacturers and grades of the materials (e.g., fungicides and reagents)
11. The detailed procedures for the treatment of plants
12. The methods to determine the phenolic contents and the data indicated in Table 3
13. Table 1 is difficult to understand the relationship between the data and others.
14. Results and discussion should also be sectionalized.
15. Page 4, line 188. XRD was used to determine/assign the crystalline structure, not to characterize.
16. Figure 1. The definition of “the sample” must be described.
17. Page 5, line 202. The peak used for the calculation of the crystallite size should be indicated.
18. Figure 2 and 3. The definitions of a-e must be indicated.
19. SEM/TEM data must be submitted to support the XRD analysis.
20. Table 3. The abbreviations are used without definition. The definitions on the superscripts are not indicated.
21. Please include some new references, such as:
Antioxidants 2022, 11(6), 1205; https://doi.org/10.3390/antiox11061205
Reviewer 2 Report
The experiments seem conducted appropriately, but the presentation is seriously flawed. I accordingly could not review the scientific validity enough. I may reconsider after the styles of this paper are sufficiently improved.
Page 2, line 51. I recommend changing the sentence to describe the global importance of viticulture. This journal is not a domestic Romanian journal, and readers of other countries should have interest on the work described.
Experimental section should be divided into appropriate sections. I could not understand the contents enough, because various matters are included without sections.
At least, I could not find the method for the synthesis of silver nanoparticles, which must be indicated.
Results and discussion should also be sectionalized.
Page 6, line 235. XRD was used to determine/assign the crystalline structure, not to characterize.
Figure 1. Are the silver nanoparticles analyzed in Figure 1 prepared using V1? The definition of “the sample” must be described.
Page 7, line 248. The peak used for the calculation of the crystallite size should be indicated.
Figure 2 and 3. The definitions of a, b, c, and d must be indicated. The explanations on the two series of the figures should be clearly indicated. I recommend separating the two series with sufficient explanation at the title. The resolution of the lower figures is too low.
The references are not fully listed and the numbers indicated are incorrect. I accordingly could not check the validity.
Reviewer 3 Report
After re-reviewing, I still cannot agree to publish this paper for the following reasons. The abstract section is too cumbersome in the presentation of the experimental material. In the introduction section, the authors did not clearly explain the relationship between the selection of D. filix-mas extract and phytosynthesized nanoparticles, which do not seem to be related to each other. If it is the killing effect on grapevine pathogens, it should also say the initial purpose of the idea. For metallic nanomaterials, the XRD data is indeed very important, but it is not at all necessary to show it with Figure1 in this work. It is more relevant to highlight the bactericidal properties of the nanomaterial suspensions. In the section of Results and discussions, the authors only show the results of the experiments, but there is little discussion and analysis. Therefore, the rigor of the conclusions cannot be determined. I suggest that this manuscript should be completely rewritten and resubmitted.